# Sugarcane-Seed-Cutting System Based on Machine Vision in Pre-Seed Mode

**DOI:** 10.3390/s22218430

**Published:** 2022-11-02

**Authors:** Da Wang, Rui Su, Yanjie Xiong, Yuwei Wang, Weiwei Wang

**Affiliations:** 1School of Engineering, Anhui Agricultural University, Hefei 230036, China; 2Anhui Province Engineering Laboratory of Intelligent Agricultural Machinery and Equipment, Hefei 230036, China

**Keywords:** sugarcane, computer vision, precision agriculture, pre-cutting mode, YOLO V5

## Abstract

China is the world’s third-largest producer of sugarcane, slightly behind Brazil and India. As an important cash crop in China, sugarcane has always been the main source of sugar, the basic strategic material. The planting method of sugarcane used in China is mainly the pre-cutting planting mode. However, there are many problems with this technology, which has a great impact on the planting quality of sugarcane. Aiming at a series of problems, such as low cutting efficiency and poor quality in the pre-cutting planting mode of sugarcane, a sugarcane-seed-cutting device was proposed, and a sugarcane-seed-cutting system based on automatic identification technology was designed. The system consists of a sugarcane-cutting platform, a seed-cutting device, a visual inspection system, and a control system. Among them, the visual inspection system adopts the YOLO V5 network model to identify and detect the eustipes of sugarcane, and the seed-cutting device is composed of a self-tensioning conveying mechanism, a reciprocating crank slider transmission mechanism, and a high-speed rotary cutting mechanism so that the cutting device can complete the cutting of sugarcane seeds of different diameters. The test shows that the recognition rate of sugarcane seed cutting is no less than 94.3%, the accuracy rate is between 94.3% and 100%, and the average accuracy is 98.2%. The bud injury rate is no higher than 3.8%, while the average cutting time of a single seed is about 0.7 s, which proves that the cutting system has a high cutting rate, recognition rate, and low injury rate. The findings of this paper have important application values for promoting the development of sugarcane pre-cutting planting mode and sugarcane planting technology.

## 1. Introduction

Sugarcane is one of the main cash crops in China [1]. Our country is a big country of sugarcane output, and sugarcane total output ranks third in the world, but at present, the degree of intelligence of the sugarcane industry in our country is still not high. Sugarcane planting and mechanization is the development trend of the industry, and the planting machine at home and abroad does not have the function of preventing injuring buds in the process of automatic cutting of cane seeds, which affects the development of the industry [2,3]. However, most of the planting machines in the world do not have the function of bud protection in the automatic cutting process of sugarcane seeds. Real-time automatic planters are usually fixed-length cutting, and the efficiency is higher than that of artificial planting [2,4,5]. The sugarcane planting technology developed in China is mainly divided into two categories according to the different cutting modes: real-time cutting planting mode and pre-cutting planting mode [6]. At present, the sugarcane planting machine used in China is mainly a real-time cutting type sugarcane planting machine, which has problems such as high labor intensity, uneven sowing, and excessive consumption of cane seeds [7]. The placement and density of the cane seeds in the pre-cutting planting mode are relatively controllable, while the germination rate and survival rate of the cane seeds are high, which has been widely promoted and applied [8,9,10,11]. To improve the quality of sugarcane seed and the efficiency of seed cutting, it needs to develop a kind of seed-cutting technology and equipment that does not harm the seed buds. So far, it has been shown that machine vision can be used to identify sugarcane stem nodes and avoid stem nodes when cutting seeds [9,12]. Foreign research on sugarcane pre-cutting machinery is ahead of China, which solves the problem of seed placement uniformity and intelligent cutting. In recent years, the research on pre-cutting machinery in China has gradually increased. Wang Sheng et al. proposed a cane seed pretreatment process that uses manipulators to transport cane seeds and cut sugarcane with a cutting knife, which is of great significance in sugarcane pre-cutting technology. The design parameters provide a certain experience for the design of cutting executing agencies [13]. Huang Yiqi et al. developed a sugarcane seed anti-damage bud-cutting system based on resistive induction counting. The system has a low rate of damaging buds, but it can only be applied to sugarcane varieties whose seed buds are raised relative to the stems of sugarcane [14]. Luo Zhiyuan et al. used single-chip microcomputer technology to control the stepper motor to achieve sugarcane cutting and found that there was a problem with a high bud injury rate during the cutting test [15]. Therefore, in the design of the sugarcane-cutting structure and seed-cutting system, the efficiency of sugarcane seed cutting should also be considered [5].

Deep learning is efficient feature extraction and object detection network structure [16,17,18,19]. Many researchers apply convolutional neural networks to agricultural research. Zhu Shiping et al. designed a wheat grain integrity image detection system based on a convolutional neural network (CNN) [20]. The system collects the images of two types of wheat grains, complete grain, and broken grain, and after segmenting and filtering the images, establishes the image database and morphological characteristic database of single grain wheat to achieve rapid and accurate identification of complete grains and broken grains of wheat grains [16]. Yang Changhui et al. built an identification and positioning system based on a convolutional neural network and KinectV2 camera. The system can obtain three-dimensional information on picking targets and obstacles and carry out obstacle avoidance picking operations in the natural environment [21]. Xia Baizhan et al. optimized the non-maximum suppression algorithm in Faster RCNN and performed a recognition test on potato sprout eyes, which improved the recognition effect of potato sprout eyes [22]. Based on YOLOv4-Tiny, Li Huipeng et al. proposed an efficient grape detection model, YOLO-grape, to solve the problem of reduced identification or recognition accuracy caused by the complex growing environments, foliage shadows and grape overlap [23]. Deep learning is more and more used in agriculture, but it is still rarely used in the identification of sugarcane eustipes. Qi Xiaokang et al. detected the litchi trunk in litchi images based on YOLOv5 as the target detection model and then extracted region of Interest (ROI) to perform semantic segmentation of litchi trunk images using the PSPNet semantic segmentation model [24]. The workflow can accurately obtain the location information of the main stem-picking point in the image of litchi. These research results put forward methods and ideas for solving the problems raised in this paper.

In this paper, a sugarcane-seed-cutting system based on deep learning technology is designed based on the characteristics of sugarcane pre-cutting planting mode and the agronomic requirements of sugarcane cutting and fully considers the cutting efficiency and germination rate. The system can carry out sugarcane-seed-cutting operations according to the needs of the planting mode, which has high cutting efficiency, and can reduce the risk of cutting-knife damage to sugarcane eustipes, so as to achieve the purpose of efficient cutting.

## 2. Materials and Methods

### 2.1. System Composition and Working Theory

In recent years, with the development of sugarcane planting technology, the advanced sugarcane pre-cutting seeding technology has also been continuously improved. The current planting technology is to use excellent sugarcane varieties to cut into sections according to requirements and then plant after cane seed screening, disinfection treatment, etc. This method is convenient for mechanized planting with the characteristics of a high germination rate, less seed usage, high yield, and low cost. The sugarcane stem leaves are stripped to leave the sugarcane stem, which is composed of an internodal segment and a stem node area. Sugarcane buds and leaf marks are found in the sugarcane node, and there is generally one bud in a sugarcane node. The sugarcane bud is located in the upper part of the stem and leaf mark near the side of the sugarcane tip. The high yield of sugarcane is closely related to seed selection, and it is better to choose the sticky tip with strong growth and full eyes. At the same time, when cutting sugarcane seeds, a 5 cm internode area should be left in the upper and lower parts of the sugarcane buds. This is because the nutrients and water required by each sugarcane bud are mainly supplied by the internodes of the bud, so a 5 cm internode can ensure sufficient nutrients and water for the germination of the sugarcane bud. Each sugarcane seed is about 12 cm long. Sugarcane-cutting agricultural requirements are shown in Figure 1.

#### 2.1.1. Sugarcane-Cutting Device Platform

The sugarcane-seed-cutting platform is the carrier of the sugarcane-seed-cutting device and is the guarantee of the normal operation of the seed-cutting system [1], and its device is shown in Figure 2. The platform is coordinated by a sugarcane conveyor, a disc knife, and a control system to complete the cutting operation. The platform is mainly composed of a cutting bench, self-tensioning conveying mechanism, reciprocating crank slider transmission mechanism, high-speed rotary cutting mechanism, DC power supply, and so on. Among them, the self-tensioning conveying mechanism includes the feeding guide rail, roller shaft, spring seat, stepper motor, etc. The reciprocating crank slider transmission mechanism includes a stepper motor, crank, connecting rod, slider, etc. The high-speed rotary cutting mechanism includes circular saw blades, DC motors, baffles, etc. The whole device adopts aluminum alloy.

#### 2.1.2. Sugarcane-Cutting System Composition

The sugarcane-cutting system is mainly composed of three parts: a visual inspection system, a control system, and a sugarcane mechanical cutting device, as shown in Figure 3. Vision inspection systems include image acquisition, image processing, and information output. When the extracted information is transmitted to the control system, it compiles the received information after self-correction and finally controls the stepper motor and DC motor (Direct Current motor) of the sugarcane mechanical cutting device to complete the transportation and cutting movement of the sugarcane. In addition, when the sugarcane eustipes detected by the vision inspection system do not meet the requirements, the information will not be sent to the control system.

#### 2.1.3. Operating Principle

The working process of cutting the sugarcane seeds is shown in Figure 4. The sugarcane is placed into the feeding guide rail and pushed inward until it is bitten by the upper and lower rollers, at which point the control system controls the *X*-axis stepper motor to drive the roller shaft to convey the seed sugarcane inward. The vision inspection system acquires the sugarcane image processing and outputs the information to the control system, which converts the distance information into a certain number of pulses of the *X*-axis stepper motor. When the *X*-axis stepper motor rotates the corresponding number of pulses, it stops and controls the *Y*-axis stepper motor to rotate a circle to drive the cutting mechanism to complete the cutting action once. In this process, the brushless DC motor drives the circular saw blade to rotate at high speed, and the speed of the brushless DC motor is changed by changing the frequency at which the governor emits PWM (Pulse Width Modulation) waves. The severed sugarcane seed slides down the discharging rail into the collection box.

### 2.2. Sugarcane-Cutting System Design

#### Seed-Cutting Device Design

(1)The Core Structure Design of the Cutting Device

The simplified model of the sugarcane-cutting device is shown in Figure 5, which can realize the *X*-axis direction of seed sugarcane transmission and the *Y*-axis reciprocating movement of the crank slider mechanism to complete the entire operation process of sugarcane cutting.

The sugarcane-cutting process consists of the *X*-axis conveying movement of the seed cane and the *Y*-axis reciprocating motion of the crank slider mechanism. The rotation of the *X*-axis stepper motor drives the belt and then drives the roller shaft of the upper and lower layers to move in the positive direction of the *X*-axis, controlling the transportation of sugarcane. The crank slider mechanism is welded into one with the rotary cutting mechanism, and the cutting of the sugarcane eustipes is achieved by the reciprocating movement of the crank slider in the positive direction of the *Y*-axis on the slide rail, and the high-speed rotation of the symmetrical circular saw blade. The two guide rods of the self-tensioning conveying mechanism slide with the rectangular guide hole, which ensures that the guide rod only moves in the horizontal direction, and the mechanism adopts a “T” rod to prevent the guide rod from detaching from the guide seat under the action of the elastic recovery force of the pressure spring. The upper and lower roller shafts are constrained by the shaft frame along the long guide groove and tend to move downwards. The overall structural design ensures the stability and versatility of the sugarcane cutting.

(2)Rotary Cutting System Design

Sugarcane stems are made up of internodal and stem regions, and there are sugarcane buds and leaf marks in the sugarcane eustipes. According to the requirements of the sugarcane-cutting process, a single sugarcane seed needs to have at least one sugarcane eustipes, and 5–10 cm long internodes on each side of the eustipes need to provide nutrients for the later growth of sugarcane seeds. As shown in Figure 6, a high-speed rotary cutting mechanism is designed according to this requirement, and the mechanism adopts a symmetrical circular saw blade with a tool adjustment groove on the blade axis. In this study, the initial distance between the two blades was designed to be 10 cm, and the adjustment range was 8–20 cm. At the same time, this distance can be adjusted to suit the length of different sugarcane seeds. In addition, since the blade needs to be in contact with sugarcane for a long time during operation, the material is selected as a 45 steel with good wearability, and its hardness is enhanced by surface heat treatment, with a thickness of 2.5 mm. There are 4 waist-shaped holes on the blade designed to reduce the resistance between the sugarcane and the blade by reducing the contact area between the two. At the same time, these 4 waist-shaped holes reduce the vibration of the blade during cutting.

When the unit is working, the circular saw blade rotates at high speed. In order to facilitate the study of the force on which sugarcane is cut, assuming that the cross-section of the cut part is a regular circle, the direction returned by the disc knife is the positive direction of the ξ axis, the vertically upward direction is the positive direction of the ψ axis, ignoring the friction in the ζ direction and the Oξψ fixed coordinate system is established with the center O of the disc knife as the origin, and the force analysis diagram is shown in Figure 7.

The mechanical equilibrium equations are listed according to the force applied to the sugarcane:(1)$$\{\begin{array}{c} F_{N1}+F_{t}cos\theta -F_{n}sin\theta =0\\ F_{N2}-G-F_{n}cos\theta -F_{t}sin\theta =0 \end{array}$$

In the equation,

$F_{t}$—the tangential force of the blade on the cutting site, N;

$F_{n}$—the positive pressure exerted by the blade on the cutting site, N;

G—sugarcane gravity, N;

$F_{N1}$—supporting force of side baffle on sugarcane, N;

$F_{N2}$—the supporting force of the ground against the cane, N;

θ—the Angle at which the blade cuts the cane, N.

(3)Communication System Design

UART communication is used between the upper-lower computers. The development board receives the TTL level signal through the on-board USB to RS232 converter connected to the USB port of the host computer, and the program converts the data into a decimal number command code with a value of 0–255, and the baud rate is set to 115,200 b/s.

### 2.3. Vision System Design

#### 2.3.1. Dataset Establishment

The sugarcane image collected by the industrial camera is pixel information, and the image is an array of M × N in the computer. As shown in Figure 8, $O_{0}$ in the image is the origin of the pixel coordinate system, and (u,v) represents the pixel abscissa and ordinate coordinates of any pixel point. The pixel coordinates of the camera lens’s optical projection point O1 are $\left(u_{0},v_{0}\right)$. Establish an image coordinate system with point O1 and x and y axes in millimeters ((x,y) is any point in the coordinate system). The width and height of each pixel in the image are $\rho_{w}$, $\rho_{h}$. v, u, and y and are calculated as follows:(2)$$v=\frac{y}{\rho_{h}}+v_{0}$$
(3)$$u=\frac{x}{\rho_{w}}+u_{0}$$
(4)$$y=\left(v-v_{0}\right)\rho_{h}$$

The correspondence between pixel coordinates and physical dimensions in the height direction is obtained by Equation (4), but due to the distortion of the lens, it is necessary to obtain the distortion parameters of the camera and then calibrate the camera to ensure that the pixel information is accurate.

In this study, an MV-SUA502C/M-T industrial camera with a focal length of 8 mm was used on the sugarcane machine bench in the Mechanical and Electrical Engineering Park of Anhui Agricultural University, and the image of sugarcane eustipes was collected on a white background, the distance from the lens to the height of sugarcane was 500 mm, the maximum resolution was 2592 × 1944, and the acquired image size was 1280 × 960. Image processing enables PyCharm2020.3. Using black-skinned sugarcane as the test material, a total of 3000 images of sugarcane under different conditions, such as different light, mud, and eustipes damage, were selected. In order to enrich the image data set, better extract the characteristics of sugarcane eustipes, and improve the generalization ability of the model, OpenCV is used to amplify the data of the original sugarcane data set. The rotation angle is randomly taken at 45° and 135°, and the original image is randomly mirrored, flipped horizontally, and flipped vertically [25,26]. Expand datasets by cropping, scaling, and more. Enhance data with image processing techniques such as adjusting saturation and hue, histogram equalization, median filtering, and more. Considering that data enhancement will lead to more serious changes in the shape and quality of images in the picture, each image is randomly amplified in the above way, and the final dataset has a total of 15,000 images [27,28].

Use the LabelImg tool to annotate the detected targets on the enhanced dataset. Considering the correspondence between labels and data and ensuring that the dataset distribution is uniform, use tools to randomly divide the dataset into training sets and test sets in a 9:1 ratio. Of these, 13,499 samples were trained set, and 1501 test set samples were stored in the format of the PASCAL VOC dataset. The test set included 10,147 images of normal sugarcane, 1552 images of damaged sugarcane, and 1800 images of sugarcane with mud [29]. The final dataset is shown in Table 1:

#### 2.3.2. Detect Network Training

This test needs to achieve rapid and accurate detection of the target eustipes of sugarcane in the natural environment; in order to achieve a rapid and robust detection effect of the vision system, the YOLO V5 model with good performance in speed and accuracy is selected.

YOLO V5 is the most recent result of the detection network based on YOLO sequence criteria. In order to obtain better training results, this paper adopts the YOLO V5s network for learning. YOLO V5s uses Mosaic data enhancement, which combines multiple images using random scaling, cropping, and arrangement. This data enhancement method is relatively good for detecting small objects because for self-defined data sets; its target recognition framework is usually to scale the size of the original image. Therefore, the anchor box and image scaling are adaptive. Its backbone uses the Focus structure and CSP structure to extract feature information from the input. CSP structure can effectively solve the gradient information repetition problem of network optimization in other large convolutional neural network framework, Backbone and integrate the gradient changes into the feature map from beginning to end. Therefore, the number of parameters and FLOPS values of the model is reduced, which not only ensures the inference speed and accuracy but also reduces the model size. The Nect part uses the same as YOLO V4, which uses PANET to aggregate features. It is based on the Mask R-CNN and FPN framework, which also enhances information dissemination. It adopts a new FPN structure with an enhanced bottom-up path, which improves the features well. The Prediction part uses GIOU Loss, which leads to faster convergence and better performance [30]. The test flow of sugarcane eustipes based on enhanced YOLO V5s is shown in Figure 9.

#### 2.3.3. Detection Information Extraction

When the sugarcane-seed-cutting system is actually working, the camera acquires the sugarcane eustipes image in real time and transmits it to the portable computer, using the detection model in the computer for real-time processing. When the vision inspection system does not detect the sugarcane eustipes target frame, the cutting instruction is not sent to the control system, and the detection of the next eustipes will then begin. As shown in Figure 10, when the visual inspection system detects the target eustipes, the target eustipes is calibrated with a red rectangular box set in advance. Set the distance between the center point of the target eustipes box and the center point of the next target eustipes box as L1, measure the distance between adjacent target boxes, in turn, L2, L3, etc., and send distance information to the control system.

When the previous cutting is completed, the host computer system converts the cutting line image coordinates according to the pixel coordinates of the cut line predicted this time and feedbacked the *X*-axis stepper motor to control the displacement of the transmission guide rail x direction. As can be seen from the formula, the two cuts before and after the adjustment displacement of the feeding guide in the x direction are calculated as follows:(5)$$L_{n+1}=\left(k_{n+1}-k_{n}\right)\rho_{h}^{\prime }$$

In order to obtain the value of $\rho_{h}^{\prime }$, the correspondence between the image pixels and the actual size needs to be established. In this paper, it can be obtained from image acquisition that the distance of 10 mm occupies 51 pixels, and the calibration coefficient $\rho_{h}^{\prime }$ = 10 mm/51 pixels in the direction of the pixel point = 0.1960 mm/pixel.

#### 2.3.4. Detect Network Comparisons

In order to verify the effectiveness and superiority of the enhanced YOLO V5s sugarcane eustipes detection network proposed in this study for cane sugarcane target eustipes recognition under complex conditions, the current representative target detection networks Faster-RCNN and YOLO V4 were trained with the same data set and training parameters, and the test set was compared and tested. In the test set, a group of common sugarcane, damaged sugarcane, and sugarcane with mud were selected for comparison and recognition, and detection. The comparison of detection results is shown in Figure 11.

According to the results, for sugarcane under normal conditions, there are a total of 6 sugarcane eustipes targets. Faster-RCNN, YOLO V4, and the enhanced YOLO V5s target detection network proposed in this paper all have higher detection rates and can identify all 6 sugarcane eustipes targets. For the damaged sugarcane, there were 6 sugarcane eustipes targets in total. The enhanced YOLO V5s target detection network could identify all the 6 sugarcane eustipes targets, while the Faster-RCNN detection network could only identify 4 of the 6 sugarcane eustipes targets, and YOLOv4 could only identify 2 of the 6 targets. Sugarcane with mud had five eustipes targets, and only the enhanced YOLO V5s could identify all of them, while both YOLO V4 and Faster-RCNN failed to identify the target of sugarcane eustipes closest to the root. When identifying the second sugarcane eustipes from the left, no target was detected by the YOLO V4 model, the IOU of the Faster-RCNN model was only 0.86, and the IOU of the enhanced YOLO V5s model was 0.99. In the whole test set, the recognition accuracy of the enhanced YOLO V5s model reaches 97.07%, which is 15.89% higher than that of the Faster-RCNN model and 8.94% higher than that of the YOLO V4 model. In conclusion, the enhanced YOLO V5s proposed in this paper significantly outperformed the other two models in the recognition rate of sugarcane eustipes targets under different conditions.

### 2.4. Control System Design

#### 2.4.1. Hardware Selection

The control system adopts STM32F407ZGT6 microcontroller as the main control chip, which is a 32-bit microprocessor developed by ST Microelectronic, whose core is ARM’s Cortex architecture, I/O port is numerous, powerful, and can meet the system design requirements. The communication method follows the standard IIC communication protocol at a frequency of 400 kHz (Max). The hardware configuration of the entire cutting system described above shows that the laptop is powered by 220 V AC, the DC motor and stepper motor are powered by 36 V, and the microcontroller is powered by 5 V. This cutting system is equipped with a 48 V lithium battery, which obtains a 5 V power supply from the lithium battery through a step-down module to power the microcontroller [31,32].

#### 2.4.2. Control Policies

The DC and stepper motors are initialized first before the sugarcane-cutting operation begins. When the control system receives the cutting instructions transmitted by the vision system and the distance information of the adjacent target frame, the distance information L1 and L2 are compiled into the number of pulses of the *X*-axis stepper motor rotation. When the *X*-axis stepper motor rotates the corresponding number of pulses, it stops and controls the *Y*-axis stepper motor to rotate a circle to drive the cutting mechanism to complete the cutting action once. During this process, the brushless DC motor always drives the circular blade to maintain a high-speed rotation. During the cutting operation, timer 2 controls the PWM and pulse output of the *X*-axis stepper motor, and timer 3 controls the PWM and pulse output of the *Y*-axis stepper motor. The core controller calls the external interrupt, timer interrupt, and other resources by controlling the enable and disable of the two timers and then controlling the non-synchronous rotation of the two stepper motors to achieve the purpose of sugarcane seed delivery cutting. The severed sugarcane seed slides down the discharging rail into the collection box. The control flow of the entire system is shown in Figure 12.

In the design of the sugarcane eustipes cutting system, the fast and smooth movement of the self-tensioning conveyor mechanism and the reciprocating crank slider conveyor mechanism are the guarantees of the quality of sugarcane cutting. Therefore, in the control of the stepper motor, a new type of fusion stepper motor control algorithm is designed. The new algorithm combines the advantages of the trapezoidal algorithm and S-type algorithm, and its core idea is to make the acceleration not mutate while the speed changes rapidly so that the controlled object can move quickly and smoothly. The new fusion stepper motor control algorithm is divided into 5 stages, namely uniform acceleration stage, varying acceleration stage, uniform speed stage, varying deceleration stage, and uniform deceleration stage. The speed control process in the new algorithm is shown in Figure 13.

As shown in Figure 14, the corresponding time of the uniform acceleration stage, varying acceleration stage, uniform speed stage, varying deceleration stage, and uniform deceleration stage are $0-t_{1}$, $t_{1}-t_{2}$, $t_{2}-t_{3}$, $t_{3}-t_{4}$,$\;\mathrm{and}\;t_{4}-t_{5}$. In order to make the transition between the velocities of movement during $t_{1}-t_{4}\;$ completely smooth, the time of the specified uniform acceleration phase is equal to the time of the deceleration phase, i.e., $t_{1}-0=t_{2}-t_{1}$. It is also stipulated that the two stages of deceleration have the same time, i.e., $t_{4}-t_{3}=t_{5}-t_{4}$. In the process of calculation, in order to make the final mathematical expression easier, it is necessary to make the time of the acceleration phase and the deceleration phase the same and set the time period to T, that is $t_{1}-0=t_{2}-t_{1}=t_{4}-t_{3}=t_{5}-t_{4}=T$. The relationship among jerk, acceleration, and velocity is:(6)$$a\left(t\right)=a\left(t_{i}\right)+\int_{t_{i}}^{t}jdt$$
(7)$$V\left(t\right)=V\left(t_{i}\right)+\int_{t_{i}}^{t}a\left(t\right)dt$$

From this, it is derived that the jerk, acceleration, and velocity mathematical models of the trapezoidal-S type acceleration and deceleration control curve are as follows:(8)$$J\left(t\right)=\{\begin{array}{c} 0,t\in \left(0,t_{1}\right)\\ -J,t\in \left(t_{1},t_{2}\right)\\ 0,t\in \left(t_{2},t_{3}\right)\\ -J,t\in \left(t_{3,}t_{4}\right)\\ 0,t\in \left(t_{4},t_{5}\right) \end{array}$$
(9)$$a=\{\begin{array}{c} a_{m},t\in \left(0,t_{1}\right)\\ a_{m}+JT-Jt,t\in \left(t_{1},t_{2}\right)\\ a_{m}-JT,t\in \left(t_{2},t_{3}\right)\\ 2JT+JT_{2}-Jt,t\in \left(t_{3},t_{4}\right)\\ -JT,t\in \left(t_{4},t_{5}\right) \end{array}$$
(10)$$V_{m}=\{\begin{array}{c} V_{0}+JTt,t\in \left(0,t_{1}\right)\\ V_{0}-\frac{1}{2}JT^{2}+2JTt-\frac{1}{2}Jt^{2},t\in \left(t_{1},t_{2}\right)\\ V_{m},t\in \left(t_{2},t_{3}\right)\\ V_{0}-\frac{1}{2}JT^{2}-2JTT-\frac{1}{2}JT_{2}^{2}+2JTt+JT_{2}t-\frac{1}{2}Jt^{2},t\in \left(t_{3},t_{4}\right)\\ V_{0}-JTt+4JT_{2}+JTT_{2},t\in \left(t_{4},t_{5}\right) \end{array}$$

In the above equations, J represents jerk; $a_{max}$ represents maximum acceleration; $V_{max}$ represents maximum velocity; and $T_{2}$ represents the time of the uniform velocity phase. It can be seen from the mathematical model that after the initial speed of the given system, the speed of each moment can be accurately calculated, and in practical applications, the establishment of a speedometer can achieve precise control of the speed of the stepper motor.

## 3. Results and Discussion

### 3.1. Test Materials and Equipment

The experimental materials are black-skinned sugarcane, subtropical herbaceous plants, and purple-black epidermis, 180–200 cm long and about 5 cm in diameter.

In order to test the feasibility and stability of the entire sugarcane-seed-cutting system, a prototype was trial-produced, and the sugarcane-cutting test was carried out in the laboratory of Anhui Agricultural University, as shown in Figure 15. The test equipment mainly includes a sugarcane-cutting machine bench, wide range programmable power supply (Hunan En-zhi Measurement and Control Technology Co., Ltd., Zhuzhou, China), 86HSE12N stepper motor (Wenzhou Pfeld Co., Ltd., Wenzhou, China), HBS86H stepper motor driver (Shenzhen Leisai Intelligent Control Co., Ltd., Shenzhen, China), 110BL DC motor (Changzhou Songjie Motor Electrical Appliance Co., Ltd., Changzhou, China), ZM-6550A DC motor driver (Changzhou Songjie Motor Electrical Appliance Co., Ltd., Changzhou, China), STM32F407ZGT6 microprocessor (Guangzhou Xingyi Electronic Technology Co., Ltd., Guangzhou, China) and sugarcane stem section cutting system. The vision inspection system adopts the MV-SUA502C/M-T camera (Shenzhen Middleway Technology Co., Ltd., Shenzhen, China), which is mounted on the cutting platform and perpendicular to the cutting plane height of 50 cm. The image processing equipment used in the test was a laptop with an Intel Core i7-9750H processor, NVIDIA GeForce GTX 1650 graphics card, and 16 GB RAM to meet basic inspection requirements.

### 3.2. Experimental Design

According to the regulations, the test takes the number of sugarcane and the actual number of eustipes as the test variables, and the identification rate of eustipes, the rate of bud injury, and the rate of cutting as the indicators for the verification test. The industrial camera works with the sugarcane-cutting system to complete the cutting.

The speed of the DC motor in the high-speed rotary cutting system is set at 3500 rpm during the test. In the trial, five black-skinned sugarcanes were randomly selected from each group for a total of 10 groups. In the experimental sugarcane samples selected, some of them had mud blocks, or the eustipes were damaged, and the top of the sugarcane that could not be used as seeds was removed. The experimental method is to record the total number of eustipes and the number of eustipes of each group of 10 sugarcane species and obtain the identification rate of eustipes ∂, the bud injury rate φ, and the cutting rate v.

Follow these steps to perform the experiment:(1)Carry out stepper motor wiring; Start the voltage regulation power supply; Adjust the DC motor speed in the high-speed rotary cutting mechanism; Open the industrial camera and the host computer.(2)Record the number of sugarcane eustipes in a group; After the motor in the working bench is stable, put a group of sugarcane into the shelf input guide in turn; Wait for all the sugarcane seed to be cut; Turn off the voltage regulation power supply, the host computer, and the industrial camera.(3)Record and calculate the eustipes recognition rate ∂, the wound bud rate φ, and the cutting rate v.(4)Repeat the above steps until all 10 groups of sugarcane have completed cutting.

### 3.3. YOLOv5 Deep Learning Vision System

The visual system was tested for sugarcane eustipes recognition using a trained model. The effectiveness of the proposed system is highlighted. Under normal circumstances, the average detection accuracy of sugarcane eustipes is 97.2%, the average detection rate of damaged sugarcane is 94.7%, and the average detection rate of muddy sugarcane eustipes is 95.9%.

Sugarcane eustipes recognition in different situations presents different challenges for the visual system. The trained model is tested to identify the sugarcane eustipes under different conditions, and the image detection effect of the sugarcane eustipes obtained under different conditions is shown in Figure 16.

### 3.4. Cutting Bench Test Results

The results of the sugarcane-cutting test of different groups are shown in Table 2. In the sugarcane-cutting test, the identification rate of sugarcane eustipes cutting is not less than 94.3%, the accuracy rate is between 94.3% and 100%, and the average accuracy is 98.2%. The rate of bud injury was measured by measuring the length of the internode zone at the ends of the sugarcane bud of the sugarcane seed. If the length of the internode region was less than 1.5 cm, we considered the seed to be substandard, from which the rate of injured buds could be derived. The injury bud rate is not higher than 3.8%. Under the action of the sugarcane self-tensioning conveying mechanism, the process of conveying sugarcane is smooth, and the time of single transportation and cutting is generally equivalent, and the average single cutting time is about 0.7 s with partial missing cutting.

Figure 17 shows the damage and bud rate of each sugarcane group and the average cutting time of each sugarcane group in the sugarcane-cutting test in Table 3. It can be seen from the trend reflected in Figure 16 that in the sugarcane-cutting test of different groups, the injury rate of sugarcane eustipes is relatively low and stable, and the injury rate is not higher than 3.8%. On the one hand, this is because the YOLO v5 network model has a high recognition rate and can well identify sugarcane eustipes. On the other hand, it is also because of the application of stepper motor algorithms. The application of stepper motor algorithms reduces mechanism jitter caused by motor start-stop.

Under the action of the sugarcane self-tensioning conveyor mechanism and stepper motor algorithm, the process of sugarcane transportation and cutting is very smooth. The entire process of sugarcane transportation and cutting is shown in Figure 18. As can be seen from the figure, the sugarcane is slowly and smoothly transported through the self-tensioning conveying mechanism, and the process is not more obvious shaking. When the camera recognizes the sugarcane eustipes, the circular saw blade begins to cut. The cutting process is relatively stable, and the average cutting time of each eustipes is relatively fixed. The average single cutting time is about 0.7 s.

### 3.5. Discussion

In this study, deep learning techniques were used in inspection systems. Compared to traditional machine vision technology, this one is more suitable for complex test conditions. Under normal circumstances, the average detection accuracy of sugarcane eustipes is 97.2%, the average detection rate of broken sugarcane is 94.7%, and the average detection rate of muddy sugarcane eustipes is 95.9%. This is because, on the one hand, the sample database of broken sugarcane and mud-stained sugarcane is small, and there are not as much sample data as normal sugarcane, which leads to a lower recognition rate of eustipes in these two types than that of normal sugarcane. On the other hand, it is also limited by the limitations of the YOLO v5 network model itself, and the identification of broken sugarcane and mud-bearing sugarcane is poor. On this basis, the laboratory will further optimize and improve the network model later.

In the sugarcane-cutting test bench experiment, we used 10 groups of black-skinned sugarcane for experiments. After analyzing the results, it was found that there were still cases where sugarcane eustipes and injured buds were not recognized, and the reasons for this were as follows. (1) The stepper motor working at a higher speed will cause the vision system to flicker and lose the target frame, thereby reducing the detection accuracy and reducing the recognition rate of the sugarcane eustipes. (2) The vision system not only needs to identify the target object but also needs to build a cutting area and issue instructions. In addition, there are many target objects in the field of view, and the high-speed movement causes the calculation of the previous frame cannot to reach the end while the data of the next frame have begun to be poured in, which increases the number of calculations required and increases the risk of error. (3) The vision system detects the target signal and transmits it to the control system. The visual system has a certain reaction time for the information transmission of the control system, and the speed is too fast to make the cutting system react in time, resulting in seed damage. According to data analysis, for normal sugarcane, while reducing the speed of the stepper motor, the vision system produces less flicker and loss of target frames. In addition, we found that the control system needed to have enough time to receive and process the information calculated by the vision system. When the speed of the stepper motor is too fast, the vision system begins to run unstable, and the detection accuracy decreases. At the same time, the response of the mechanical structure lags behind, so the cutting strategy constantly fails and restarts. This is consistent with our expected results. In view of these problems, we will further optimize the algorithm in the subsequent research, and we will test with different varieties of sugarcane to ensure the reliability of the system.

## 4. Conclusions

(1) The sugarcane-seed-cutting system, including a visual inspection system, control system, and mechanical seed-cutting device, is designed to meet the cutting operations of sugarcane in different types and different environments. The system operates stably and can complete a series of processes such as sugarcane transportation, sugarcane eustipes identification, positioning, and cutting.

(2) A sugarcane-seed-cutting mode based on multi-motor coordination is proposed. In order to realize this movement mode, the cutting device is designed, and the reciprocating motion of the sugarcane conveying and cutting mechanism is completed by 2 stepper motors, the cutting of sugarcane is completed by 1 DC motor, and the rapid cutting of sugarcane eustipes can be realized with the cooperation of 3 motors, and the cutting action is rapid.

(3) A visual system based on the YOLO V5 model was established to detect sugarcane eustipes, and the single frame image processing time was reduced by constructing a high-pixel scale feature layer, and the recognition rate of sugarcane eustipes was as high as 94.3% during the test, which verified the feasibility of the sugarcane eustipes visual detection system.

(4) In the cutting system, the stepper motor precision control algorithm is adopted, which makes the process of sugarcane transportation and cutting more stable and further protects the eustipes of sugarcane.

(5) The test tested the recognition rate, bud injury rate, and cutting rate of sugarcane seed of different groups of sugarcane, and the test results showed that the recognition rate of sugarcane eustipes was not less than 94.3%, the accuracy rate was between 94.3% and 100%, and the average accuracy rate was 98.2%. The injury bud rate is not higher than 3.8%, the average cutting time of a single eustipes is about 0.7 s, and compared to manual, the cutting system has a higher cutting rate, recognition rate, and low bud injury rate.

## Figures and Tables

**Figure 1 sensors-22-08430-f001:**
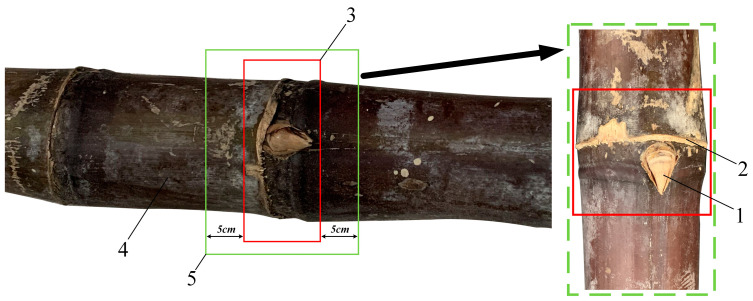
Sugarcane-cutting agricultural requirements: 1—sugarcane buds; 2—leaf scar; 3—nodes area; 4—internodes area; 5—a sugarcane seed.

**Figure 2 sensors-22-08430-f002:**
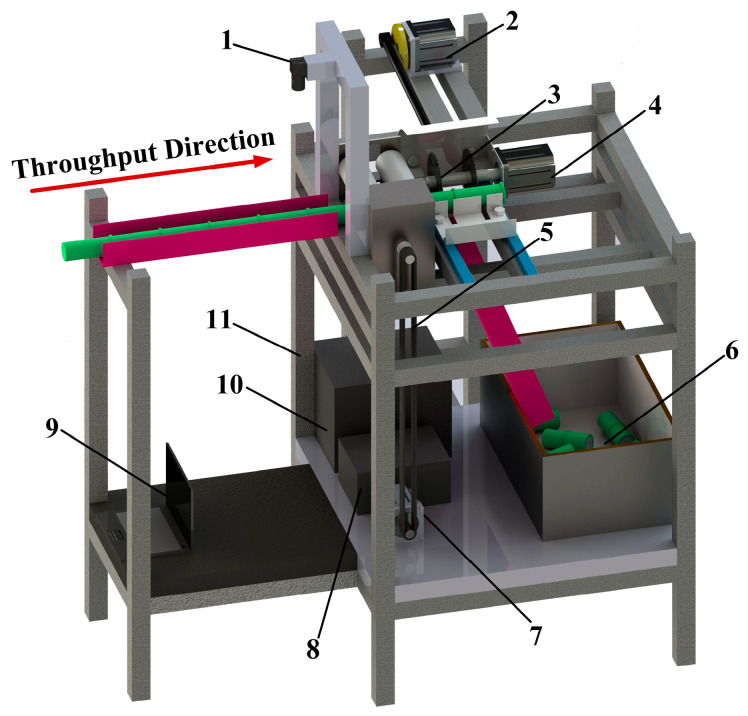
Schematic diagram of the platform structure of the sugarcane-cutting device: 1—Monocular camera; 2—*X*-axis stepper motor; 3—Circular saw blades; 4—DC motor; 5—Belt; 6—Discharge guide; 7—*Y*-axis stepper motor; 8—Battery; 9—Computer; 10—Electronic control part; 11—Bench.

**Figure 3 sensors-22-08430-f003:**
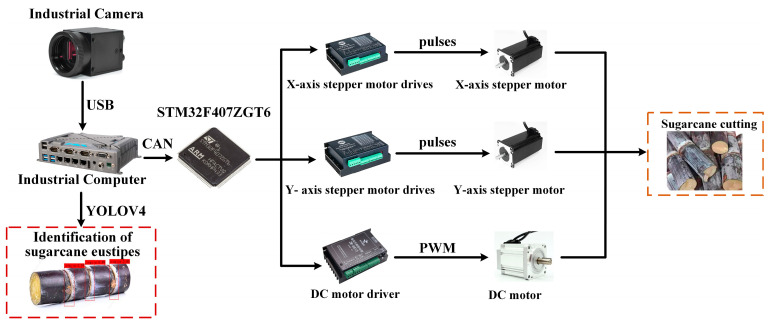
Block diagram of the seed-cutting system.

**Figure 4 sensors-22-08430-f004:**
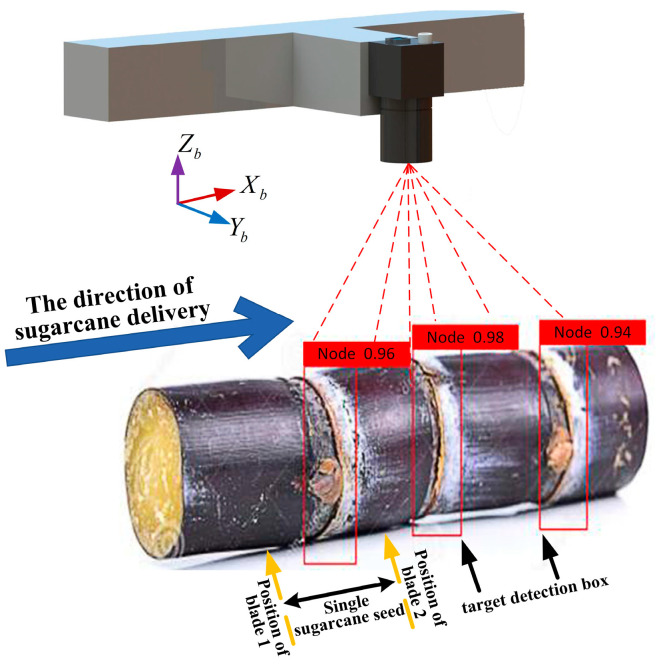
Schematic diagram of sugarcane-cutting patterns.

**Figure 5 sensors-22-08430-f005:**
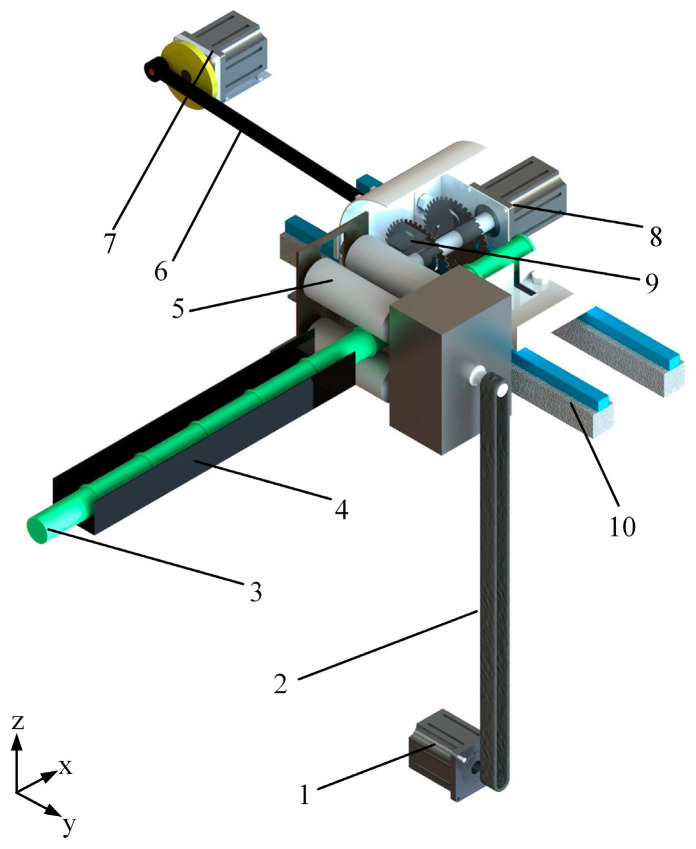
Model of the core structure of the cutting device: 1—*X*-axis stepper motor; 2—Belt; 3—Sugarcane; 4—Guide rails for feeding; 5—Roller; 6—Crank connecting rod mechanism; 7—*Y*-axis stepper motor; 8—DC motor; 9—Circular saw blades; 10—Slide.

**Figure 6 sensors-22-08430-f006:**
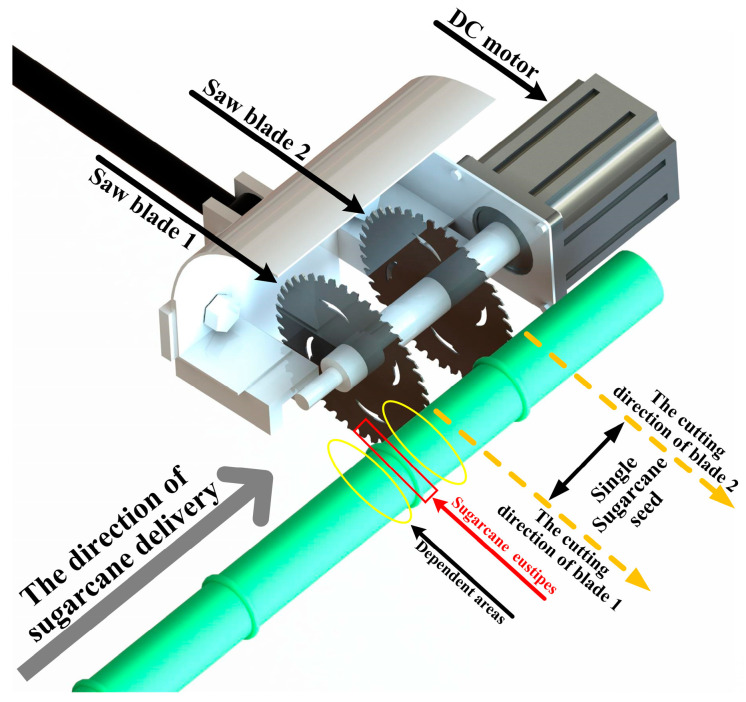
Schematic diagram of the working blade of the cutting blade.

**Figure 7 sensors-22-08430-f007:**
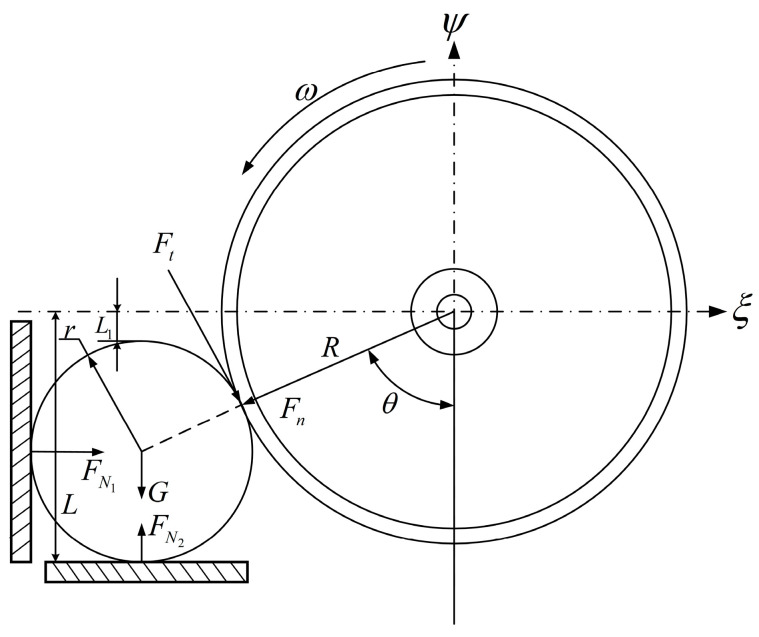
Static stress analysis of sugarcane cutting.

**Figure 8 sensors-22-08430-f008:**
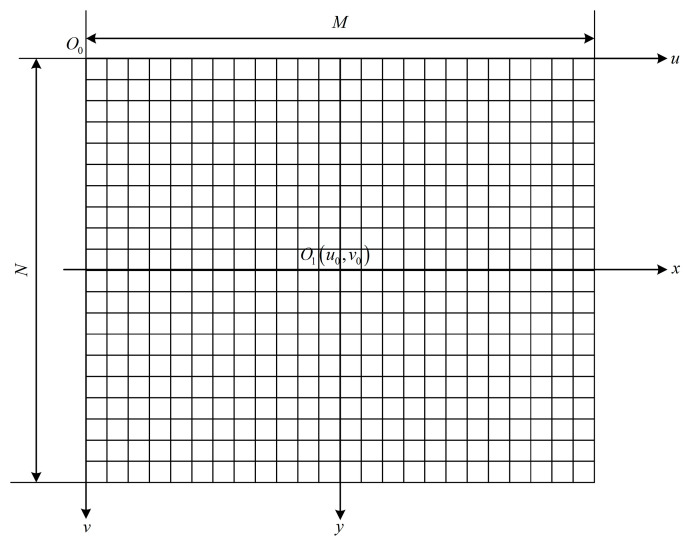
Image coordinate system.

**Figure 9 sensors-22-08430-f009:**
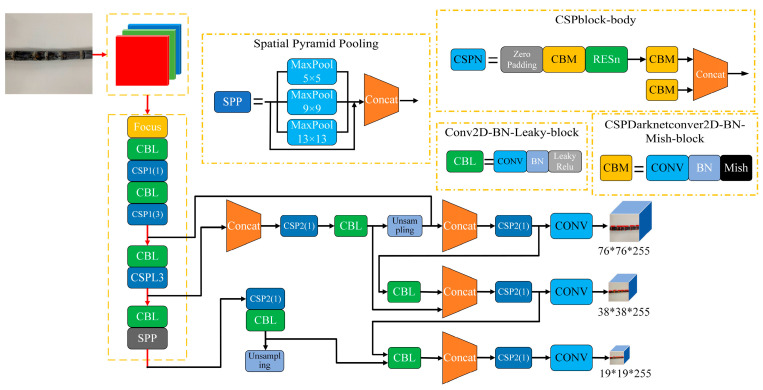
Detection of sugarcane target eustipes based on YOLO V5s.

**Figure 10 sensors-22-08430-f010:**
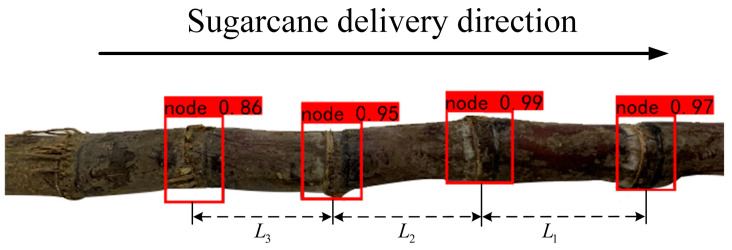
Schematic of vision system inspection.

**Figure 11 sensors-22-08430-f011:**
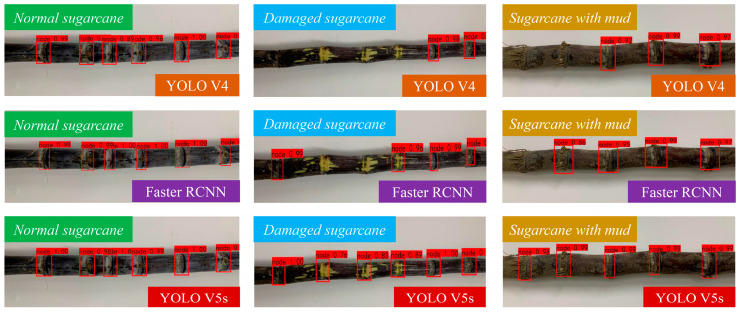
Sugarcane target recognition under different models.

**Figure 12 sensors-22-08430-f012:**
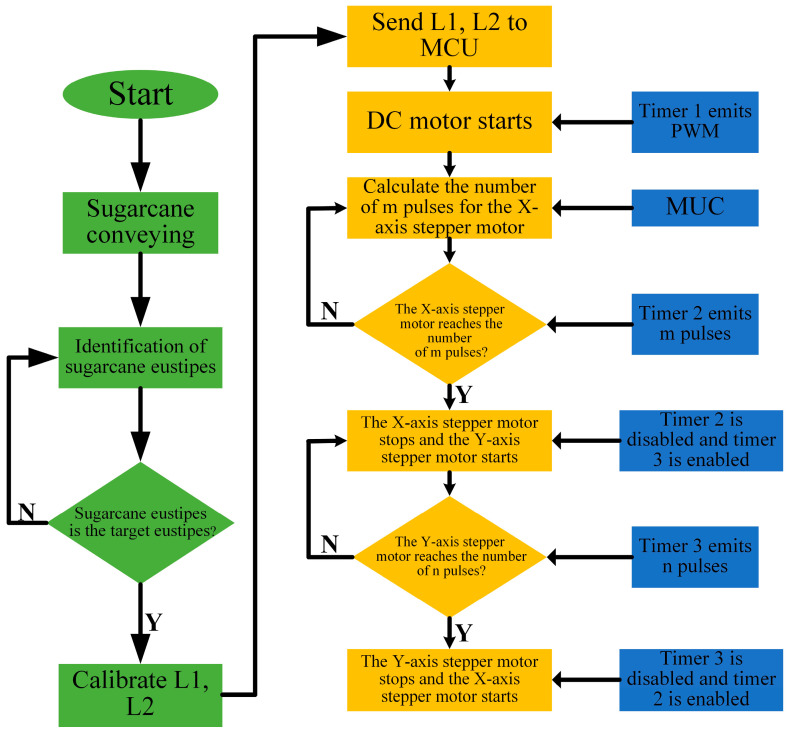
System control flowchart.

**Figure 13 sensors-22-08430-f013:**
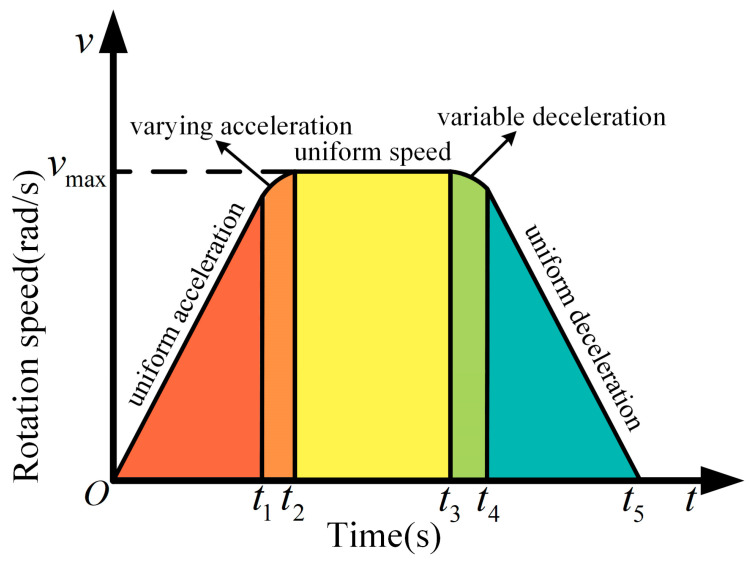
Speed control model diagram. $0-t_{1}$ is the uniform acceleration stage, $t_{1}-t_{2}$ is the varying acceleration stage, $t_{2}-t_{3}$ is the uniform speed stage, $t_{3}-t_{4}$ is the varying deceleration stage, $t_{4}-t_{5}$ is the uniform deceleration stage.

**Figure 14 sensors-22-08430-f014:**
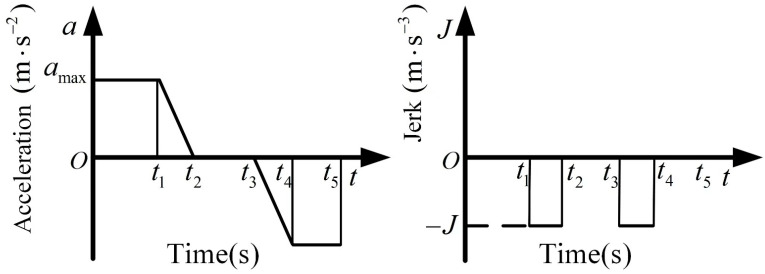
Acceleration and jerk algorithm model diagram.

**Figure 15 sensors-22-08430-f015:**
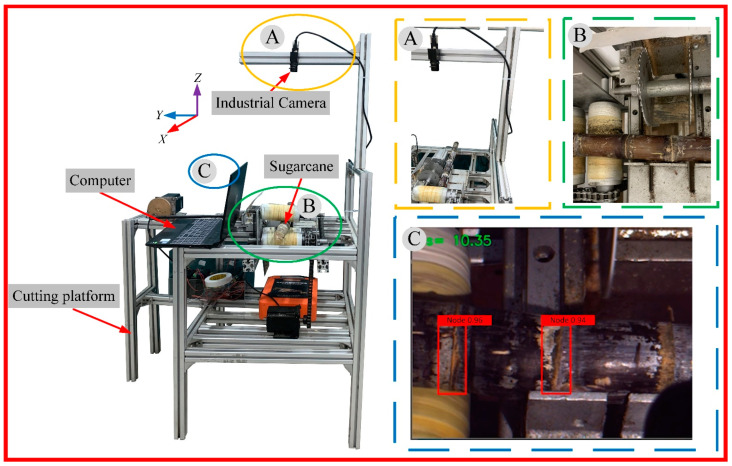
Laboratory test of sugarcane-seed-cutting device. ((**A**): Enlarged view of the mounting position of the industrial camera; (**B**): Partial magnification of the sugarcane cutting device; (**C**): Screenshot of the computer running interface).

**Figure 16 sensors-22-08430-f016:**
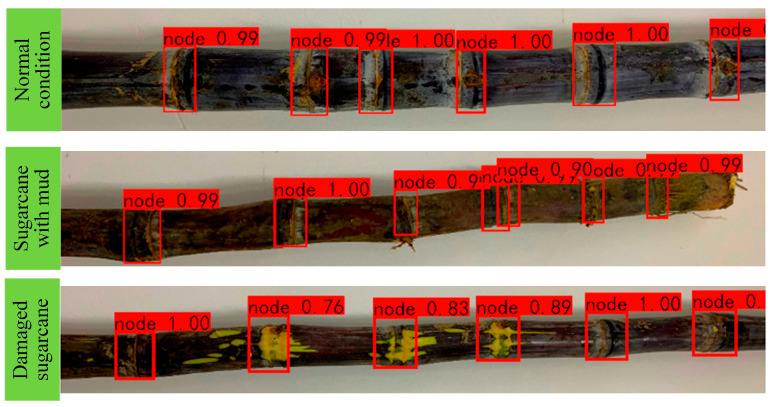
Identification renderings of sugarcane eustipes under different conditions.

**Figure 17 sensors-22-08430-f017:**
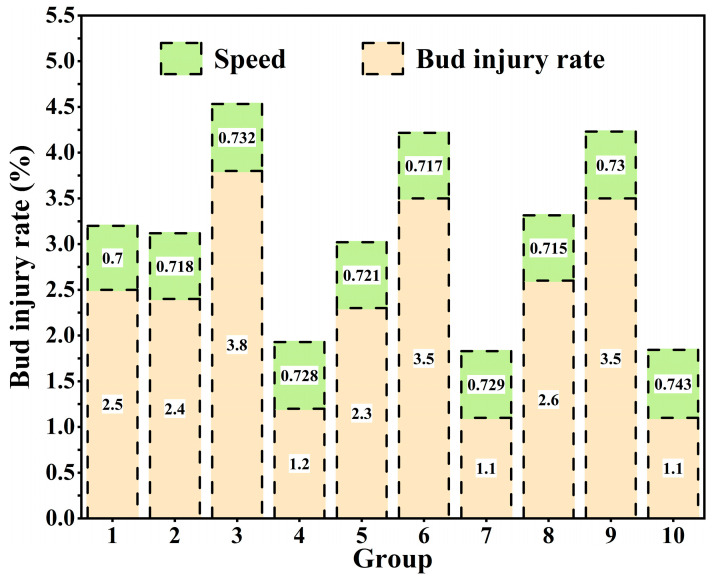
Analysis of test data results.

**Figure 18 sensors-22-08430-f018:**
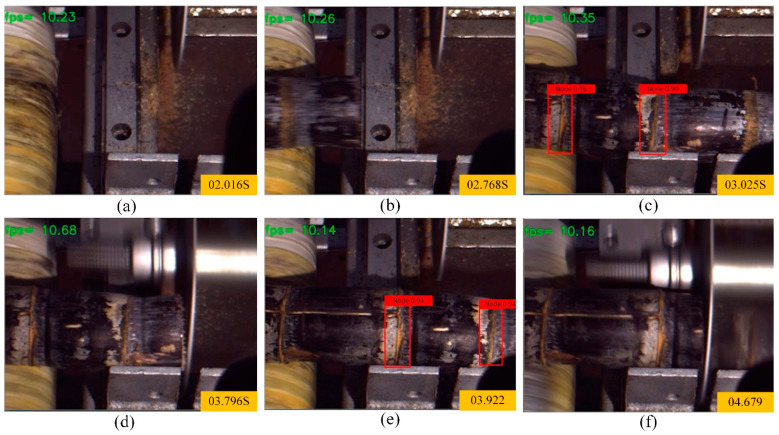
Dynamic process of sugarcane cutting ((**a**–**f**) represent the images captured by the computer of sugarcane node recognition at corresponding time).

**Table 1 sensors-22-08430-t001:** The number of images after data enhancement.

Data Type	Normal Type of Sugarcane	Damaged Sugarcane	Sugarcane with Mud
Training set	10,147	1552	1800
Test set	1128	173	200
Total	11,275	1725	2000

**Table 2 sensors-22-08430-t002:** Comparison of the recognition rate and injury rate of eustipes and the rate of damaged buds in different groups of sugarcane.

Number of Groups	Actual Number of Eustipes/pcs	Number of Eustipes Cut/pcs	Rate of Eustipes Recognition/%	Number of Buds Injured/pcs	Rate of Buds Injured/%
1	79	78	98.7	2	2.5
2	83	83	100.0	2	2.4
3	78	76	97.4	3	3.8
4	81	80	98.8	1	1.2
5	85	85	100.0	2	2.3
6	85	83	94.3	3	3.5
7	88	86	97.7	1	1.1
8	77	75	97.4	2	2.6
9	85	84	98.8	3	3.5
10	89	88	98.9	1	1.1

**Table 3 sensors-22-08430-t003:** Comparison of seed cutting speed o in different groups of sugarcane.

Number of Groups	Total Number of Eustipes/pcs	Number of Eustipes Cut/pcs	Number of Eustipes Cut/pcs	Cutting Time/s
1	18	18	12.6	0.700
2	17	17	12.2	0.718
3	20	19	13.9	0.732
4	18	18	13.1	0.728
5	19	19	13.7	0.721
6	19	18	12.9	0.717
7	17	17	12.4	0.729
8	21	20	14.3	0.715
9	20	20	14.6	0.730
10	23	21	15.6	0.743

## Data Availability

The data used in this study are self-tested and self-collected. As the control method designed in this paper is still being further improved, data cannot be shared at present.

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
