# Peer review of "Sugarcane-Seed-Cutting System Based on Machine Vision in Pre-Seed Mode"

_sensors, 2022, doi:10.3390/s22218430_

Round 1
Reviewer 1 Report
The paper deals with an updated problem and applies an enhanced solution for sugar cane cutting. The central issue of the manuscript is that it is in its present form. It is a technical report and not a scientific paper. There are communicated such technical details that are straightforward engineering solutions for the tasks related to the cutting machines. For example, synchronizing the cutting subsystem with the image vision is necessary in such cases. If one part's speed is much higher than the other, the solution might be the parallelization of the slower subsystem. Also, the image vision system probably has a ready signal, then it starts the conveying process, so we need to reduce the conveyor speed, but as fast as possible, the cane should be transported for the cutting phase. But all these comments relate to the technical details that must be omitted from a manuscript making a scientific paper instead of an engineering report.
The introduction contains citations the reviewer could not identify with the list of references; for example, Wang Sheng is not the author of [12] since J. Ali is. Same problems are with 12, 13, 18, 19, 20, etc. Check the proper author names and numbers for all citations.
Figure 1: 1 and 2 are better to indicate in the enlarged subpicture since it is more precise than the major one, and also, the red frame should appear in the subpicture.
Cutting system design:
In that chapter, widely known engineering solutions are applied and mentioned. These are the natural components of a design process. Generally, they are not treated in scientific papers.
According to the Sensor scope, the image vision system might interest the journal readers. The reviewer believes the paper must be shortened, focusing on the image vision problems and solutions as a scientific solution. The manuscript should deal with the issues scientifically and not as a technical report.
In L237: There is mentioned equation 6, but this is a mistake; equation 4 is probably needed.
Figure 10: Numbering the nodes is unclear for the reader, probably some type.
The belt accelerating – decelerating process part (L360-366) is confusing since the time equations contain some errors; check them.
Line 480-486: This part is confusing. There is no image inspection ready signal? If it is, the problem that is treated here is unclear.
Author Response
Dear peer reviewers and editors
Hello! Thank you very much for your professional and wise comments on this article. According to the experts' questions and opinions, the author gives a detailed explanation in the form of one question and one answer, and marks the manuscript with blue font in the corresponding position, as follows:
QUESTION1: The introduction contains citations the reviewer could not identify with the list of references; for example, Wang Sheng is not the author of [12] since J. Ali is. Same problems are with 12, 13, 18, 19, 20, etc. Check the proper author names and numbers for all citations.
RE: Thank you very much for your valuable comments. We apologize for our negligence. We carefully checked and checked the numbers and names of all documents in the text according to your suggestions. At the same time, we have standardized the expression and citation of articles Chinese literature.
QUESTION2: Figure 1: 1 and 2 are better to indicate in the enlarged subpicture since it is more precise than the major one, and also, the red frame should appear in the subpicture.
RE: Thank you very much for your valuable comments. We think your advice is pertinent. We modified Figure 1 and placed 1 and 2 and the red frame in the subpicture.
QUESTION3: Cutting system design: In that chapter, widely known engineering solutions are applied and mentioned. These are the natural components of a design process. Generally, they are not treated in scientific papers.
RE: Thank you very much for your valuable comments. The innovation point of this article is to propose a sugarcane cutting scheme under the pre-cut seed planting mode, and design a sugarcane cutting device. We consider the design of the sugarcane cutting system and the detailed description of the plant to be an important part of this article. There are also appropriate innovations in the design of the device. So we believe that the contents of this part can be retained.
QUESTION4: According to the Sensor scope, the image vision system might interest the journal readers. The reviewer believes the paper must be shortened, focusing on the image vision problems and solutions as a scientific solution. The manuscript should deal with the issues scientifically and not as a technical report.
RE: Thank you very much for your valuable comments. We believe that the manuscript should deal with the issues scientifically and not as a technical report and some of the content in the cutting system design in the whole text, such as the hardware design of the cutting system, is superfluous, and it does not have much effect on the focus of the whole text. So we've scaled down the full text to better highlight the point.
QUESTION5: In L237: There is mentioned equation 6, but this is a mistake; equation 4 is probably needed.
RE: Thank you very much for your valuable comments. We apologize for the mistakes we made. We have made changes in the manuscript.
QUESTION6: Figure 10: Numbering the nodes is unclear for the reader, probably some type.
RE: Thank you very much for your valuable comments. Due to the distance between sugarcane stem nodes, the identification labels are unclear. We have re-tested the sugarcane identification test to avoid occlusion.
QUESTION7: The belt accelerating – decelerating process part (L360-366) is confusing since the time equations contain some errors; check them.
RE: Thank you very much for your valuable comments. We checked the contents of this section. To simplify the mathematical expression, we specify that the two acceleration stages and the two deceleration phases have the same time. So there is the expression presented in the article.
QUESTION8: Line 480-486: This part is confusing. There is no image inspection ready signal? If it is, the problem that is treated here is unclear.
RE: Thank you very much for your valuable comments. In an image inspection system, we have an image detection ready signal. When two target boxes are identified, a distance signal is generated and transmitted to the controller. The content of our section is mainly an analysis of the causes of errors in the identification process of sugarcane stem nodes. We discuss these factors that may affect detection accuracy in detail and make further improvements in subsequent studies.
Thank you very much for your valuable opinions. Please review the revised article!

Reviewer 2 Report
Sugarcane seed cutting system based on YOLO V5 in pre-seed mode
This paper discussed an application of deep-learning or 4.0 technology for an agricultural industry where it aimed to develop a new intelligent sugarcane seed-cutting system based on deep learning in 12 pre-seed planting modes for identification and cutting of sugarcane target eustipes.
Overall this work presented an interesting discussion. Nevertheless, several parts need to be improved and added to the current manuscript.
1. Introduction regarding sugarcane industry needs to be presented by statistical data. How significant exactly the demand to develop the discussed technology must be stated in the manuscript.
2. There is no comprehensive review regarding specific coding for the deployed deep learning in this work.
3. Following comment no 2, it seems like this work only selected several instruments for detection, and then deployed it this research to achieve the designated purpose. In this case, overall design process to produce the presented platform of sugarcane cutting (Fig. 2) also has to be discussed in this work. How is such platform is designed? Where is technical drawing of the platform?
4. Details of instruments are not complete. Make sure to put series, manufacturer, city, and country.
5. Check how the number citation in the text is written. Make sure to put space between last word and bracket for citation.
6. Which variant of sugarcane is suitable for this technology? Since in my fast screening, at least more thant five varieties of sugarcane are exist. It is also noted that authors mentioned "10 sugarcane species", but no tables or list what kind of species varieties deployed for this work.
7. Equations have not been clarified. Each symbol and mathematical writing must be clarified.
8. Make sure to clarify every abbreviation.
9. Is there any docummentation of the experimental process (instalation, main process, and post-processing)? It is required to support the conducted set-up of the laboratory experiment.
10. The discussion is too straightforward and details of this work in methodology is missing. Even though this work is intended to sensor-based research, but it also involve subject of agriculture and engineering design. These aspects need to be significantly improved. Authors are also requested to add discussion regarding improvement methodology of this work. How results in Table 2 and Figure 16 can be a base-line to achieve better outcome.
11. Results of the experiment are highly recommended to be calculated again using statistical method so that trendline regarding which is the best species (match with the platform function) can be concluded, etc.
12. State the recommendation for future works in refereces.
13. Revise format of the references according to journal guidelines.
Author Response
Dear peer reviewers and editors
Hello! Thank you very much for your professional and wise comments on this article. According to the experts' questions and opinions, the author gives a detailed explanation in the form of one question and one answer, and marks the manuscript with blue font in the corresponding position, as follows:
QUESTIONS1: Introduction regarding sugarcane industry needs to be presented by statistical data. How significant exactly the demand to develop the discussed technology must be stated in the manuscript.
RE: Thank you very much for your valuable comments. We further elaborated our country in the introduction of the article sugarcane industry scale and proportion in the world. This paper proves the necessity of this research by analyzing the market demand.
QUESTIONS2: There is no comprehensive review regarding specific coding for the deployed deep learning in this work.
RE: Thank you very much for your valuable comments. The code for the YOLO V5 network model is now open source. We applied the YOLO V5 network model to the engineering practice of sugarcane seed cutting. In this paper, we introduce the network model in detail, and introduce the operation process of the whole system. The feasibility of the network model can be seen from the experimental results. We have detailed code for this study, but we do not think it is necessary to show this in the article.
QUESTIONS3: Following comment no 2, it seems like this work only selected several instruments for detection, and then deployed it this research to achieve the designated purpose. In this case, overall design process to produce the presented platform of sugarcane cutting (Fig. 2) also has to be discussed in this work. How is such platform is designed? Where is technical drawing of the platform?
RE: Thank you very much for your valuable comments. The platform of the sugarcane seed cutting device is introduced in detail in line 114 to 124 of the article. This includes the composition and name of each part of the device, the equipment used, etc. In fact, Picture 2 is a 3D drawing of the sugarcane seed cutting device designed by us. The 3D drawing is in strict agreement with the physical drawing of the device we designed. The feasibility of the drawing design has also been fully verified in the actual test.
QUESTIONS4: Details of instruments are not complete. Make sure to put series, manufacturer, city, and country.
RE: Thank you very much for your valuable comments. We should pay attention to this problem. However, based on the valuable comments of other reviewers, we believe that the introduction of hardware in the paper is too much, which makes the paper look like a technical report rather than a research paper. So we reduced the hardware part of the article to better address these issues.
QUESTIONS5: Check how the number citation in the text is written. Make sure to put space between last word and bracket for citation.
RE: Thank you very much for your valuable comments. We have checked the citations throughout the text to ensure that the format is normative and accurate.
QUESTIONS6: Which variant of sugarcane is suitable for this technology? Since in my fast screening, at least more than five varieties of sugarcane are exist. It is also noted that authors mentioned "10 sugarcane species", but no tables or list what kind of species varieties deployed for this work.
RE: Thank you very much for your valuable comments. According to the current actual situation, the experiment in this paper uses black cane, and our experiment is also carried out based on black cane. In the subsequent research, we will conduct experimental research on other varieties of sugarcane. The "10" mentioned in the text means that we carried out 10 groups of experiments using black cane instead of using 10 varieties of cane.
QUESTIONS7: Equations have not been clarified. Each symbol and mathematical writing must be clarified.
RE: Thank you very much for your valuable comments. We have clarified and explained every symbol in the equation.
QUESTIONS8: Make sure to clarify every abbreviation.
RE: Thank you very much for your valuable comments. We went through a careful check to make sure that each abbreviation was explained in detail.
QUESTIONS9: Is there any documentation of the experimental process (installation, main process, and post-processing)? It is required to support the conducted set-up of the laboratory experiment.
RE: Thank you very much for your valuable comments. We introduced the whole process and steps of sugarcane seed cutting experiment in detail. At the same time, the device and equipment used in the test are also described in the test section.
QUESTIONS10: The discussion is too straightforward and details of this work in methodology is missing. Even though this work is intended to sensor-based research, but it also involve subject of agriculture and engineering design. These aspects need to be significantly improved. Authors are also requested to add discussion regarding improvement methodology of this work. How results in Table 2 and Figure 16 can be a base-line to achieve better outcome.
RE: Thank you very much for your valuable comments. We improved the discussion part of the trial. According to the results in Figure 16 and Table 2, we have analyzed the causes of errors and the causes of damage and bud. In view of these problems, we will further optimize the algorithm in the subsequent research. And we will test with different varieties of sugarcane to ensure the reliability of the system.
QUESTIONS11: Results of the experiment are highly recommended to be calculated again using statistical method so that trendline regarding which is the best species (match with the platform function) can be concluded, etc.
RE: Thank you very much for your valuable comments. We check and recalculate the statistical results and present them visually. Conclusions can be drawn on the basis of the test results.
QUESTIONS12: State the recommendation for future works in references.
RE: Thank you very much for your valuable comments. We improved and checked the references to make sure they were normative. In the discussion, we also further elaborate our subsequent research work.
QUESTIONS13: Revise format of the references according to journal guidelines.
RE: Thank you very much for your valuable comments. We downloaded the Sensor Journal reference format and modified the format of all our references.
Thank you very much for your valuable opinions. Please review the revised article!

Reviewer 3 Report
Well done work. Supported by appropriate tables, graphs and pictures. The formula described in the calculation is good. The derivation is good. The English language is selective. Possibly more references could have been made.
Author Response
Dear peer reviewers and editors
Hello! Thank you very much for your professional and wise comments on this article. We have corrected the deficiencies you pointed out in the manuscript. We have polished the English of the article and added some references.
Thank you very much for your valuable opinions. Please review the revised article!

Reviewer 4 Report
Dear Authors,
I rate the work very highly. Here are some small remarks.
1. Please check picture 7 for:
- that all forces are correctly described,
- whether they have the correct phrases,
- whether they are all included in the drawing.
When cutting through the material, a frictional force is generated.
2. Lack of explanation of the markings used in Figure 7.
3. I don't understand the structure of Figure 16 - it needs to be changed or clarified.
Kind regards
Author Response
Dear peer reviewers and editors
Hello! Thank you very much for your professional and wise comments on this article. According to the experts' questions and opinions, the author gives a detailed explanation in the form of one question and one answer, and marks the manuscript with blue font in the corresponding position, as follows:
QUESTION1: Please check picture 7 for: that all forces are correctly described, whether they have the correct phrases, whether they are all included in the drawing. When cutting through the material, a frictional force is generated.
RE: Thank you very much for your valuable comments. We checked the force of picture 7 and further standardized the expression. We think of all the forces in the X and Y axes as being represented in this diagram. For the friction caused by the cutting process, we consider its direction to be the Z-axis direction perpendicular to the X-axis and Y-axis, which has little influence on the direction of our force analysis, so we ignore it in the figure.
QUESTION2: Lack of explanation of the markings used in Figure 7.
RE: Thank you very much for your valuable comments. We have explained the notation in detail at the back of the formula.
QUESTION3: I don't understand the structure of Figure 16 - it needs to be changed or clarified.
RE: Thank you very much for your valuable comments. Figure 16 shows the damage and bud rate of each sugarcane group and the aver-age cutting time of each sugarcane group in the sugarcane cutting test. The lower bar chart shows the cut bud rate of each group of sugarcane, and the upper bar chart shows the average cut time. We've gone a step further in this article.
Thank you very much for your valuable opinions. Please review the revised article!

Reviewer 5 Report
Comments and suggestions for authors and will be shown to the authors
● The paper states that sugarcane is one of the main cash crops in China, and its planning and mechanization is the development trend of the automatic cutting of cane seeds
● Such machines technology has been developed in two modes: 1) Real-time cutting mode and 2) pre-cutting planting mode
● The sugarcane planting machines used in China are mainly a real-time sugarcane cutting mode type, which has problems of high labor intensity, uneven sowing and excessive consumption of cane seeds.
● These machines are made so that the sugarcane seed buds at the cane nodes are kept safe and unharmed, which needs to identify stem nodes using machine vision, mainly in the pre-cutting mode machines
● In the design of sugarcane cutting structure and seed cutting system, the efficiency of sugarcane seed cutting should also be considered.
● Deep learning is an efficient feature extraction and object detection network structure and has been used in agriculture research in identifying the broken and complete wheat grains. The authors claim to have been using Deep learning in the identification of sugarcane eustipes.
● The paper describes a deep learning-based sugarcane cutting system based on the pre-cutting planting mode characteristics and the agronomic requirements of sugarcane cutting, and fully considers the cutting efficiency and germination rate.
Query 1. The agronomic requirement should be explained and show how the cutting efficiency is being measured, possibly in the form of germination rate as results?
Is the subject matter presented in a comprehensive manner?
● The paper has explained system composition and bud cutting theory according to requirement after sugarcane leaves stripping off the stem plant for seed screening and disinfection treatment as shown in Figure 1.
● The sugarcane cutting device is as shown in Figure 2, consisting of a sugarcane conveyor belt, disc knife, and a control system with DC motors which are powered by a DC battery to complete the cutting operation.
● The sugarcane cutting system is further explained with working principle in Figure 3 and the operating principle in Figure 4, consisting of vision inspection system to output the vision information to the cutting control system operated by brushless stepper motors (X-axis and Y-axis motors shown in Figure 5) with PWM-based speed control
● The Schematic diagram of the working blade of the cutting blade is shown in Figure 6 and Static stress analysis of sugarcane cutting is given in Figure 7 supported by mechanical equations with image coordination given in Figure 8
Query 2. The paper looks more descriptive, descripting either the process of machines or giving images for explanation of the YOLO V5.
Query 3. The authors have not been able to maintain a proper flow to convince readers about the measurable results in conclusion.
Query 4. The detection of sugarcane target eustipes based on YOLO V5 is given in Figure 9 and schematic of vision system inspection is given in Figure 10, but showing no
Are the references provided applicable and sufficient?
The authors take support from thirty one (31) current and branded references including some from MDPI. Some references are of mechanical scope and not even properly formatted, for example, Reference 15, Reference 30 and Reference 31 are reproduced as under:
- Li, Xiang, Qian Ding, and Jian-Qiao Sun. "Remaining useful life estimation in prognostics using deep convolution neural networks." Reliability Engineering & System Safety 172 (2018): 1-11.
- Xue, Hongtao, Ziwei Song, Meng Wu, Ning Sun, and Huaqing Wang. "Intelligent Diagnosis Based on Double-Optimized Artificial Hydrocarbon Networks for Mechanical Faults of In-Wheel Motor." Sensors 22, no. 16 (2022): 6316.
- Buzzy, Michael, Vaishnavi Thesma, Mohammadreza Davoodi, and Javad Mohammadpour Velni. "Real-time plant leaf counting using deep object detection networks." Sensors 20, no. 23 (2020): 6896.
Query 5. The references are properly formatted and new relevant references are added.

Author Response
Dear peer reviewers and editors
Hello! Thank you very much for your professional and wise comments on this article. According to the experts' questions and opinions, the author gives a detailed explanation in the form of one question and one answer, and marks the manuscript with blue font in the corresponding position, as follows:
Query 1: The agronomic requirement should be explained and show how the cutting efficiency is being measured, possibly in the form of germination rate as results?
RE: Thank you very much for your valuable comments. In Line 95 to 107, we describe the agronomic requirements for sugarcane seed cutting. The sugarcane bud is located in the upper part of the stem and leaf mark near the side of the sugarcane tip. Generally speaking, a cane seed has only one stem node, and there are 5 cm long internodes on each side of the stem node to provide nutrients for the late stage of the sugarcane seeds. Sugarcane cutting agricultural requirements are shown as Figure 1. For the cutting efficiency, we prepared the sugarcane used for the test in advance and put it in turn through the feeding guide rail to start the timing. When all the sugarcane has been cut, we count the number of sugarcane seeds, we can get the time to cut a sugarcane seed. This is an important basis to measure the stability and feasibility of cutting system and cutting device. Since it is difficult to measure the germination rate of sugarcane, we take the damage bud rate as the basis for judging the test results. We adopt the standard recognized by the industry, when the distance between the cutting end face of sugarcane and sugarcane bud is less than a value, it is unqualified sugarcane seed, so as to obtain the damage bud rate we need.
Query 2: The paper looks more descriptive, descripting either the process of machines or giving images for explanation of the YOLO V5.
RE: Thank you very much for your valuable comments. The innovation of this article lies in the proposed method of sugarcane seed cutting under the pre-cut planting mode. In order to improve the quality of sugarcane cut seeds, we used the YOLO V5 network model to identify the stem segments of sugarcane. At the same time, we developed and designed a sugarcane seed cutting device based on this model. The whole text is also around these two innovation points. This paper first puts forward the device of sugarcane seed cutting, and then introduces the internal identification method.
Query 3: The authors have not been able to maintain a proper flow to convince readers about the measurable results in conclusion.
RE: Thank you very much for your valuable comments. For Conclusion 1, In the introduction of cutting system device, we have introduced the sugarcane cutting device in detail. The design of the device ensures the stability of sugarcane cutting. In order to achieve accurate recognition of sugarcane stem segments, we used YOLO V5 network model to identify sugarcane stem segments. According to the final test process and test results, the reliability of conclusion 1 is also illustrated. For Conclusion 2, This is an innovation in our device design. We use multi-motor coordination to cut sugarcane, which is rare. According to the cutting efficiency of the cutting test, the design of this device is good. For Conclusion 3, We used the current mainstream YOLO V5 network model to identify sugarcane. The feasibility of the network model can be seen from the results of sugarcane identification test and cutting test. For Conclusion 4, Aiming at the problem of motor start-stop jitter in sugarcane cutting process, we adopted S-type stepper motor algorithm. It can be seen from the average cutting time that the whole cutting process is very stable without obvious jitter. For Conclusion 5, This is a summary of the results of the trial data. This further confirms the reliability of the first four conclusions.
Query 4: The detection of sugarcane target eustipes based on YOLO V5 is given in Figure 9 and schematic of vision system inspection is given in Figure 10, but showing no
RE: Thank you very much for your valuable comments. But we can't see all of your questions. Figure 10 is the result of sugarcane identification based on the network model proposed in Figure 9. At the same time, in Figure 9, we also show the results of sugarcane images after the network model.
Query 5: The references are properly formatted and new relevant references are added.
RE: We have made appropriate changes and revisions to the literature. We carefully checked the numbers and names of all documents in the text according to your suggestions. At the same time, we have standardized the expression and citation of articles Chinese literature.
Thank you very much for your valuable opinions. Please review the revised article!

Reviewer 6 Report
I have attached file to this email.

Author Response
Dear peer reviewers and editors
Hello! Thank you very much for your professional and wise comments on this article. According to the experts' questions and opinions, the author gives a detailed explanation in the form of one question and one answer, and marks the manuscript with blue font in the corresponding position, as follows:
QUESTION1: There are several types as well as grammatical errors which make the text difficult to understand on many occasions.
RE: Thank you very much for your valuable comments. The grammar of the entire manuscript has been carefully examined and the English has been polished.
QUESTION2:Did the authors compare the results of their study with the work of other authors?
RE: Thank you very much for your valuable comments. We compared with the current research status of sugarcane seed cutting. For sugarcane seed cutting, the current research focuses on Li Shang-ping's team from Guangxi University and Fan Yun-lei's team from Jiangnan University. In contrast, the cutting system we designed has some innovation points. First of all, we used YOLO V5 network model to identify sugarcane stem segments, which improved the recognition rate of sugarcane stem segments and reduced the bud damage rate of sugarcane cutting. The second is the innovation of sugarcane seed cutting device. We designed a self-tensioning conveying mechanism and a high speed rotary cutting mechanism to ensure the practicability and efficiency of the sugarcane seed cutter. Finally, we added the stepper motor S type algorithm in the motor control to ensure the stepper motor start and stop smoothly, so as to further ensure the health of sugarcane seeds and the germination rate of sugarcane. So based on that we think our research is very innovative and practical. The research results have far-reaching significance for promoting the development of sugarcane pre-cut planting mode.
QUESTION3: The parameters for the axes are not clear in Figure 10.
RE: Thank you very much for your valuable comments. We apologize for the mistakes we made. We have made changes in the manuscript.
QUESTION4: The parameters for the axes are not clear in Figure 11.
RE: Thank you very much for your valuable comments. We apologize for the mistakes we made. We have made changes in the manuscript.
QUESTION5: The conclusion is brief and does not cover all aspects of work Some important findings can be presented in Abstract.
RE: Thank you very much for your valuable comments. The conclusions of this study are restated in the paper, and our work is summarized in more detail in the abstract6
QUESTION6: Some important findings can be presented in Abstract.
RE: Thank you very much for your valuable comments. We redescribe some of our important findings in the abstract.
QUESTION7: The current references are appropriate. However, the reference list is short. Consequently, some new references about the research subject should be added to the reference list and cited in the text such.
RE: Thank you very much for your valuable comments. We have added some new references to the research topic and cited them in the text.
Thank you very much for your valuable opinions. Please review the revised article!

Round 2
Reviewer 1 Report
The modifications that the authors made improved the manuscript, but some basic mistakes remained or were generated:
Figure 1:
Check the list, at least 5 is not a sugarcane seed, but 1 is. The numbers seems to be in reverse order.
Figure 2:
X and Y direction is confusing. Check it! X is the feeding Y is the cutting according to the later text, as in line 148-150.
Figure 7:
In this figure X and Y are local coordinates and not correspond to the whole system. It should be mentioned or using a different letter, might be Greek type.
Line 380-388:
As it was said, the time ranges are confusing. The ranges have negative values in applied definitions.
Consider instead of 0 - t1 -> t1-0, t2-t1, etc. (greater minus lower).
Author Response
Dear reviewer, the author has revised your questions one by one. Please refer to the attachment for details. Thank you again for your modification. I wish you good health and all the best!

Reviewer 2 Report
All questions have been addressed in the revision.
Author Response
Dear peer reviewers and editors
Thank you very much for your professional and wise comments on this article. Your comments on the manuscript have been very helpful in improving the quality of the manuscript. After revising the manuscript, you have acknowledged the quality of our work. Thanks to your professional standards, the quality of the entire manuscript has been improved, for which we are deeply grateful.
Wish you good health and good work.
Reviewer 5 Report
Pls see the attachment

Author Response

(The authors gave the same response as above.)

Reviewer 6 Report
The Authors have correctly made all the required changes.
Author Response

(The authors gave the same response as above.)
